# DOPA Homeostasis by Dopamine: A Control-Theoretic View

**DOI:** 10.3390/ijms222312862

**Published:** 2021-11-28

**Authors:** Rune Kleppe, Qaiser Waheed, Peter Ruoff

**Affiliations:** 1Norwegian Center for Maritime and Diving Medicine, Haukeland University Hospital, 5021 Bergen, Norway; rune.kleppe@helse-bergen.no; 2Department of Chemistry, Bioscience and Environmental Engineering, University of Stavanger, 4021 Stavanger, Norway; qaiser.waheed@uis.no

**Keywords:** dopamine, DOPA, tyrosine, homeostasis, metabolic channeling, robust control, integral feedback, derepression, tyrosine hydroxylase, neurotransmitter, vesicles, mathematical modeling, zero-order kinetics, oxidative stress, Parkinson’s disease

## Abstract

Dopamine (DA) is an important signal mediator in the brain as well as in the periphery. The term “dopamine homeostasis” occasionally found in the literature refers to the fact that abnormal DA levels can be associated with a variety of neuropsychiatric disorders. An analysis of the negative feedback inhibition of tyrosine hydroxylase (TH) by DA indicates, with support from the experimental data, that the TH-DA negative feedback loop has developed to exhibit 3,4-dihydroxyphenylalanine (DOPA) homeostasis by using DA as a derepression regulator. DA levels generally decline when DOPA is removed, for example, by increased oxidative stress. Robust DOPA regulation by DA further implies that maximum vesicular DA levels are established, which appear necessary for a reliable translation of neural activity into a corresponding chemical transmitter signal. An uncontrolled continuous rise (windup) in DA occurs when Levodopa treatment exceeds a critical dose. Increased oxidative stress leads to the successive breakdown of DOPA homeostasis and to a corresponding reduction in DA levels. To keep DOPA regulation robust, the vesicular DA loading requires close to zero-order kinetics combined with a sufficiently high compensatory flux provided by TH. The protection of DOPA and DA due to a channeling complex is discussed.

## 1. Introduction

Dopamine (DA) is an important neurotransmitter. Its dysfunction is associated with a variety of neuropsychiatric disorders. Low activity levels of DA are related to movement disorders, irregular sleep patterns, or attention deficit hyperacticity disorder (ADHD). On the other hand, increased or dysregulated DA activity in different brain areas have been related to addiction [1] and schizophrenia [2,3] but also to improved cognitive functions [4]. Thus, the literature often refers to DA homeostasis in order to address the need for optimum DA levels to assure proper nervous functions.

The enzyme tyrosine hydroxylase (TH) converts tyrosine (Tyr) to the precursor 3,4-dihydroxy-phenylalanine (DOPA), which becomes DA by decarboxylation using dopa decarboxylase (DDC) (Figure 1). Cellular DA is moved into vesicles by vesicular monoamine transporter 2 (VMAT2); the vesicles are transported to the active site possibly by kinesins [5]. There, vesicles are emptied into the synaptic cleft upon the arrival of action potentials and the inflow of Ca${}^{2+}$ ions [6]. There is evidence that the reaction chain Tyr → DOPA → vesicular DA inside the presynaptic cell is organized in terms of “metabolite channeling” [7], where the enzymes TH, DDC, and the transport VMAT2 form a complex attached to the vesicular membrane [8,9,10,11,12]. In the synaptic cleft DA is spread by diffusion and by reuptake into the presynaptic terminals with DA transporters (DAT) or by uptake by neighboring glial cells where it is metabolized [13,14,15].

There are several regulatory interactions which act on TH [23,31,32]. In addition, DOPA and dopaquinone have been reported to act as inhibitors [23,33], although the TH inhibition by quinones appears to be related to irreversible modifications of TH’s sulfhydryl groups [33]. In addition, TH has been found to possess DOPA oxidase activity [34] similar to the enzyme Tyrosinase (TYR) [28,29]. TYR, though expressed at low levels in dopaminergic neurons, uses DOPA as a substrate and seems to form neuromelanin [35].

DA synthesis and metabolism have been modeled in several studies addressing different regulatory aspects and model approaches. One of the earlier models of DA synthesis, release, and metabolism used the assumption that DOPA-derived DA enters the vesicular pool without mixing with cytosolic DA [36]. Three DA pools were considered, i.e., inactive bound DA, releasable bound DA, and free cytosolic DA. The TH regulation of DA synthesis was modeled [37], with an emphasis on regulation by phosphorylation/dephosphorylation, the cofactor BH4, and the cell’s redox status. Multiple activity states of TH were considered [37] together with influences by iron and α-synuclein. These enzymatic models are based on Michaelis–Menten (MM) kinetics. Others models described DA synthesis and metabolism by a power-law approach (biochemical systems theory) developed by Savageau [38]. The calculations by Qi et al. [30,39,40] contain an analysis of presynaptic DA homeostasis including details on DA metabolites, catechol auto-oxidation, and melanin formation. Best et al. [27] modeled DA synthesis and metabolism including TH regulation by autoreceptors. Cullen and Wong-Li [41] analyzed a reduced version of the model by Best et al. [27], which is considered to be computationally more efficient and addressing more clearly underlying key mechanisms. Véronneau-Veilleux et al. [42] developed an integrative model for Parkinson’s disease by integrating a pharmacokinetic Levodopa/DOPA treatment with DA dynamics and a neurocomputational model of basal ganglia. This illustrates nicely how different models can be combined to bridge interactions between different subsystems and to obtain clinically relevant predictions. Muddapu et al. [43,44] used a multiscale approach to model a network between the brain regions Substantia Nigra Pars Compacta (SNc), Subthalamic Nucleus (STN), Globus Pallidus externa (GPe ) with a spiking neuron model. To study the loss of dopaminergic cells [44] and Levodopa-induced toxicity [45], the Muddapu et al. model contains details about ion channels, calcium buffering mechanisms, energy metabolism pathways, DA turnover processes, and calcium-induced apoptosis. Recently, a detailed single-cell model of SNc was used to study the influence of energy deficiency on its subcellular processes to obtain insights into the neurodegeneration in Parkinson’s disease [46]. Practically all modeling approaches are based on solving the rate equations in the form of ordinary differential equations (ODEs). Among the most detailed models in terms of metabolites are probably the studies by Qi et al. [30].

In the present study, we use a minimal model approach to understand possible key (and ideal) features of DA regulation, which might not be immediately visible from a more complex model. Focus is taken on the presynaptic TH-DA negative feedback loop from a robust homeostatic viewpoint. As described in more detail below, we show that low $K_{M}$ values for the transport of DA into vesicles (as indicated by experimental data) allow for robust homeostasis of DOPA by using DA as the regulator. An intact homeostasis of DOPA by DA leads, in addition, to a maximum DA loading of vesicles. The feedback regulation of DOPA by DA is explored with respect to influences of oxidative stress, aging, and Levodopa treatment on DOPA and DA levels.

### Mechanisms Leading to Robust Homeostasis

Before we go into more detailed properties of the TH-DA negative feedback and to DOPA regulation, we describe, by taking a simple negative feedback loop as an example, the requirements for robust homeostasis.

The term homeostasis was introduced by Walter Cannon in 1929 as “the sum of mechanisms which keep the steady states in organisms within certain but narrow limits” [47,48]. With the development of cybernetics [49,50,51] and systems biology [52,53,54], feedback mechanisms have become central in our understanding of how homeostasis can be achieved in biological systems [55,56].

During the past 20 years, evidence has accumulated that organisms can employ regulation methods, which originally were discovered/invented in control engineering [57,58,59,60,61,62,63,64,65,66,67,68]. Integral control, as one of these engineering methods, was invented in the beginning of the 20th century for the robust steering of ships [69,70] but is now an essential element in practically all industrial control applications [57].

In integral control, the difference between the controlled variable A and its set-point $A_{set}$ is integrated (Figure 2). The integrated error ϵ is proportional to the concentration of the manipulated (controller) variable, which is used to oppose perturbations of the controlled variable A. It can be proven that for step-wise perturbations the controlled variable A always converges toward its set-point $A_{set}$ [57]. This is generally referred to as the ”robustness” of the controller.

How can error integration be performed in a biochemical system? There are presently three approaches to achieve integral control in reaction kinetics: one is based on zero-order kinetics in the removal of the manipulated variable E [58,60], while another is based on a first-order autocatalytic formation of E combined with its first-order removal kinetics [65,71,72]. A third approach uses so-called antithetic control, where two controller variables are involved [65,66,68]. We focus here on how robust control can be achieved by zero-order kinetics.

Figure 3 gives a simple example of a negative feedback loop between *A* and *E*. Component *P* is a precursor of the controlled variable *A*. This feedback loop can be considered as a minimum model of TH inhibition by DA where *P*, *A*, and *E* are, respectively, Tyr, DOPA, and DA.

The rate equations are:(1)$$\frac{dP}{dt}=k_{1}-\left(\frac{k_{2}\cdot P}{k_{3}+P}\right)\cdot \left(\frac{k_{4}}{k_{4}+E}\right)-k_{10}\cdot P$$
(2)$$\frac{dA}{dt}=\left(\frac{k_{2}\cdot P}{k_{3}+P}\right)\cdot \left(\frac{k_{4}}{k_{4}+E}\right)-\underset{\mathrm{perturbation}}{\underset{︸}{k_{9}\cdot A}}-\left(\frac{k_{5}\cdot A}{k_{6}+A}\right)$$
(3)$$\frac{dE}{dt}=\left(\frac{k_{5}\cdot A}{k_{6}+A}\right)-\left(\frac{k_{7}\cdot E}{k_{8}+E}\right)$$

The rate equation for *E* includes an MM-type transport/removal term $k_{7}\cdot E/\left(k_{8}+E\right)$. When this term becomes zero order with respect to *E*, i.e., $k_{8}\ll E$ such that $E/(k_{8}+E)\approx \;1$, the steady-state value of *A* is under robust control with set-point $A_{set}$:(4)$$\frac{dE}{dt}=\left(\frac{k_{5}\cdot A}{k_{6}+A}\right)-k_{7}\overset{!}{=}0\;\;\;\Rightarrow \;\;\;A_{ss}=A_{set}=\frac{k_{6}k_{7}}{k_{5}-k_{7}}$$

In relation to integral control (Figure 2), we can rewrite the rate equation for *E* in the following form (see Appendix A):(5)$$\frac{dE}{dt}=\gamma_{A}\left(A_{set}-A\right)$$
with $\gamma_{A}=-\left(k_{5}-k_{7}\right)⁄\left(k_{6}+A\right)$. $\left(A_{set}-A\right)$ represents the error between the actual value of *A* and $A_{set}$. However, unlike to the presentations of integral control for linear or simpler chemical models [57,64]; here, $\gamma_{A}$ is not a constant but depends upon A.

To maintain robust control in *A* the flux $j_{2}\;\left(=k_{2}P/\left(k_{3}+P\right)\right)$ needs to be large enough to be able to compensate for the perturbing flux $j_{9}=k_{9}\cdot A$; otherwise, the concentrations of *A* are below $A_{set}$. This is illustrated in Figure 4, where step-wise perturbations by $k_{9}$ are applied during four phases. In the third phase, $j_{2}$ is too low to compensate for the increased perturbation ($k_{9}$ = 5.0). In this phase, *E* is practically zero, and $j_{2}$ cannot be increased further by *E*-depression. However, a larger *P* concentration in phase 4 by an increased $k_{1}$ value allows one to reestablish homeostasis in *A*. E is increased and can again act as a derepressing regulator *E* and compensate for the outflow perturbation $j_{9}=k_{9}\cdot A$.

## 2. Materials and Methods

### 2.1. Method of Calculation

Rate equations were solved by using the Fortran subroutine LSODE [73]. Plots and curve fits were generated with gnuplot (version 5.4.2, www.gnuplot.info) and edited with Adobe Illustrator CS6 (version 16.0.0, adobe.com). To make notations simpler, concentrations of compounds are generally denoted by compound names without square brackets. Concentrations of the Tyr → DOPA → DA reaction chain are given in μM, and time units are in minutes.

### 2.2. Model of Tyrosine Hydroxylase Regulation

Figure 5 shows the considered model:

The DAergic midbrain neurons form highly extended and complex structures, in particular, their axonal arbors [79]. We can therefore expect considerable compartmentalization in these neurons. Our modeling approach does however not rely on compartmentalization or model this explicitly. Although metabolite intermediates are free in solution and act through mass action kinetics, we expect that local concentration differences of the metabolites can occur within the axonal structure. We tentatively assume that the enzymes TH, DDC and the transporter VMAT2 form a channeling complex as indicated in Figure 1. When using the term ’cytosolic DOPA’ or ’cytosolic DA’ later in the paper, we do not necessarily consider that DOPA and DA are homogeneously distributed in the cytoplasm, but due to the local distribution of the channeling complex at the vesicular membrane, we assume that the highest concentrations of free DA and DOPA are physically close to the complex (this situation is termed by Cornish-Bowden “dynamic channeling”; see Figure 13.30 in [7]). Despite the additional inhibitions acting on TH, the key element for a possible robust homeostasis is here the same as for the model in Figure 3, i.e., the occurrence of zero-order kinetics in the removal of the TH-inhibiting species DA. Figure 5 shows that this step is related to the loading of DA into the vesicles by transporter VMAT2. DOPA is thereby the potentially regulated species, and DA is the regulator. The transport of DA into the vesicles is ATP dependent, which is needed to build a necessary proton gradient by ATPase [10,80]. The ATP dependence is not included in the model.

Experimentally determined rate parameters and inhibition constants (see next sections) are used and include substrate (Tyr) inhibition of TH [24], the inhibition of TH by DA [22,23,31], and the inhibition of TH by extracellular DA via D2 autoreceptors [27]. Furthermore, DOPA is enzymatically removed both by TH [34] as well as by TYR. The experimentally determined $K_{M}$ value for the transport of DA into vesicles, which is critical for a robust regulation of DOPA by DA has indeed been found to be relatively low (0.29 μM [81]; see also [82] for another low $K_{M}$ value with respect to vesicle loading). These $K_{M}$ values indicate that the TH-DA negative feedback may prove to be close to robust behavior.

ROS, such as OH${}^{\cdot }$ radicals or superoxide O${}_{2}^{-}$, play an additional factor for removing DOPA (leading to neuromelanin [30]), thereby perturbing DOPA homeostasis. The influence of external DOPA addition by mimicking Levodopa medication (with rate constant $k_{22}$) is included in the model together with the re-entry of extracellular DA into the cell. More detailed descriptions of the experimental parameter/rate constant values are given in the next sections.

The rate equations are:(6)$$\frac{d(Tyr)}{dt}=k_{1}-\left(\frac{k_{2}\cdot Tyr}{k_{3}\left(1+\frac{DA}{k_{4}}\right)\cdot \left(1+\frac{DOPA}{k_{16}}\right)+Tyr}\right)\left(\frac{k_{12}}{k_{12}+Tyr}\right)\left(\frac{k_{20}}{k_{20}+DA_{ex}}\right)-k_{10}\cdot Tyr$$
(7)$$\begin{array}{cc} \frac{d(DOPA)}{dt} & =\left(\frac{k_{2}\cdot Tyr}{k_{3}\left(1+\frac{DA}{k_{4}}\right)\cdot \left(1+\frac{DOPA}{k_{16}}\right)+Tyr}\right)\left(\frac{k_{12}}{k_{12}+Tyr}\right)\left(\frac{k_{20}}{k_{20}+DA_{ex}}\right)\\ & -\frac{k_{9}\cdot DOPA}{k_{13}+DOPA}-\frac{k_{5}\cdot DOPA}{k_{6}+DOPA}-k_{14}\cdot DOPA-\frac{k_{15}\cdot DOPA}{k_{16}\left(1+\frac{Tyr}{k_{3}}\right)+DOPA}+k_{22} \end{array}$$
(8)$$\frac{d(DA)}{dt}=\frac{k_{11}\cdot DA_{ex}}{k_{21}+DA_{ex}}+\frac{k_{5}\cdot DOPA}{k_{6}+DOPA}-\frac{k_{7}\cdot DA}{k_{8}+DA}-\alpha \cdot \frac{k_{18}\cdot DA}{k_{19}+DA}+k_{23}\cdot \left(DA_{ves}\right)$$
(9)$$\frac{d(DA_{ves})}{dt}=\frac{k_{7}\cdot DA}{k_{8}+DA}-k_{17}\cdot DA_{ves}-k_{23}\cdot \left(DA_{ves}\right)$$
(10)$$\frac{d(DA_{ex})}{dt}=k_{17}\cdot DA_{ves}-\frac{k_{18}\cdot DA_{ex}}{k_{19}+DA_{ex}}$$

The grayed term in Equation (8) indicates the removal of DA at increased MAO concentrations with α being an adjustable factor. The set-point for DOPA ($DOPA_{set}$) is calculated, in an analogous manner as for the P-A-E model (Figure 3), i.e., from the steady-state condition of Equation (8) with α = 0. The influence of α is shown below when considering combination aging, oxidative stress, and DA auto-oxidation.

In calculating the set-point $DOPA_{set}$, we consider that the transport velocity of DA into the vesicles, given by $k_{7}\cdot DA/\left(k_{8}+DA\right)$, is an ideal zero-order process with respect to DA. In other words, at low enough $k_{8}$ values, we have that $k_{7}\cdot DA/\left(k_{8}+DA\right)\approx k_{7}$.

Then, inserting the DA re-entry flux
(11)$$j_{11}=\frac{k_{11}\cdot DA_{ex}}{k_{21}+DA_{ex}}$$
and the vesicular DA leakage (for details see next section and Appendix B)
(12)$$j_{23}=k_{23}\cdot \left(DA_{ves}\right)$$
into Equation (8), setting d(DA)/dt=0, and solving for DOPA, gives the expression for the DOPA set-point:(13)$$DOPA_{set}=\frac{k_{6}\left(k_{7}-j_{11}-j_{23}\right)}{k_{5}+j_{11}+j_{23}-k_{7}}$$

Due to the re-entry and leakage terms $j_{11}$ and $j_{23}$, $DOPA_{set}$ depends on the DA level in the synaptic cleft and on the amount of DA stored in vesicles. The maximum set-point of DOPA is reached when $j_{11}$ = 0 and $j_{23}$ = 0 with $DOPA_{set}^{max}=k_{6}k_{7}/\left(k_{5}-k_{7}\right)$. $DOPA_{set}$ goes to zero when fluxes $j_{11}$ and $j_{23}$ balance $k_{7}$, i.e., when there is no net-uptake of DA into vesicles. Thus, under ideal controller conditions (low $k_{8}$), $DOPA_{set}$ reflects the net-uptake of DA into vesicles.

### 2.3. Parameter Values

To the extent these were found in the literature, experimentally determined rate parameter values were used in the calculations. While reported $K_{M}$ values can be directly applied, $V_{max}$ values depend on the actual enzyme concentrations, which are mostly unknown. Rate constants (see the scheme of the model, Figure 5) $k_{1}$, $k_{10}$, $k_{11}$, $k_{14}$, $k_{17}$–$k_{20}$, and $k_{22}$ were used as adjustable parameters. In the following section, we give an overview of found the literature data.

#### 2.3.1. Dopamine Transport into Vesicles and Leakage

Robust homeostasis of DOPA is based on zero-order kinetics for the transport of DA into vesicles (i.e., a low transporter $K_{M}$ value compared to the local DA concentration transporter VMAT2 is exposed to). Volz et al. [81] studied the DA uptake kinetics by VMAT2 using voltammetry. The $V_{max}$ and $K_{M}$ values were determined, respectively, to 1.9 fmol/(s·mg protein) and 0.289 μM. Experimental determination of cytosolic DA levels were measured by capillary electrophoresis and voltammetry with values between 2 and 3 μM [83,84,85] and 47–140 μM [86]. Near [82] studied the binding of [2-3H]Dihydrotetrabenazine to bovine striatal synaptic vesicles. The kinetic determination of the equilibrium constant resulted in the value of 5.4 nM ($5.4\times 10^{-3}\;\mu$M), which relates to the $K_{M}$ for vesicle loading under a rapid-equilibrium assumption.

There is evidence that catecholamine transmitters stored in vesicles leak into the cytoplasm [16,17,18,19,20]. An analysis of the kinetic data by Fried [16] and Schonn et al. [17] (Appendix B) show that the vesicular leakage of DA can be described as a first-order process with respect to $DA_{ves}$ (Equation (12)). While both Fried and Schonn et al. data sets give $k_{23}$ values close to each other, we use $k_{23}$ = (0.0158 ± 0.0006) min${}^{-1}$ from the Schonn et al. data.

#### 2.3.2. Dopamine Re-Entry by DAT

DA transport by DAT was reported to follow Michaelis–Menten kinetics with $K_{M}$ estimates varying between 0.2 to 2 μM ($k_{21}$ in the model) [77,78]. In our calculations, we treated the re-entry of DA by DAT as a zero-order process (low $k_{21}$). Although different $K_{M}$ values of DAT change $DOPA_{set}$ by changed $j_{11}$ fluxes (as the leakage flux $j_{23}$ of DA out from vesicles also does (see Equation (13)), we found that different $DOPA_{set}$ values due to different DAT $K_{M}$’s are still defended.

#### 2.3.3. DOPA Decarboxylase (DCC) EC 4.1.1.28

BRENDA [87] lists $K_{M}$ values for human DDC of 0.028 [88] and 4.27 mM [89]. We take the multiplicative average of these rather different values and use $k_{6}=\sqrt{0.028\cdot 4.27}$ mM = 346 μM. The turnover number of the enzyme was determined to be 5.1 s${}^{-1}$ or 306 min${}^{-1}$ [88]. Assuming (rather arbitrarily) a DCC concentration of 10${}^{-8}$ M, we obtain a $V_{max}$ ($k_{5}$) value of 3.06 μM/min, which is used in the calculations.

#### 2.3.4. Tyrosinase (TYR) EC 1.14.18.1

Three human $K_{M}$ entries for DOPA as a substrate are reported: 0.34 [28], and 0.48 and 0.49 mM [29]. We take the average and assign for $k_{13}$ a value of 437 μM. The turnover number for this enzyme is reported to be 38.11 s${}^{-1}$ or 2286 min${}^{-1}$ [28]. Using an enzyme concentration of 10${}^{-9}$ M, the used $V_{max}$ ($k_{9}$) value in the calculations is 2.3 μM/min.

#### 2.3.5. Tyrosine Hydroxylase (TH) EC 1.14.16.2. Inhibition by Tyr

BRENDA has 5 entries for the inhibition constant $k_{12}$ of TH by Tyrosine. These are data of four mutant enzymes and one wild-type enzyme with the values [24]: 0.037, 0.044, 0.046 (wild type), 0.048, and 0.073 mM. The average value is 0.0496 mM. Fossbakk [25] reports a similar inhibition constant as [24] for the wild-type enzyme (59 ± 18 μM) and, in addition, for a variety of TH point mutations. In the calculations, we use a value of 50 μM for $k_{12}$.

#### 2.3.6. Inhibition of TH by DA

Using bovine TH, Lazar et al. [22] reports an inhibition (dissociation) constant for DA, $k_{4}$ in our model, in dependence of the enzyme’s phosphorylation status. At pH 7.0, the inhibition constant is 6 ± 2 μM for the nonphosphorylated form and 45 ± 7 μM when the enzyme is phosphorylated. Sura et al. [23] report for human TH isoforms inhibition constants for DA in the range 78–208 μM for S40 phosphorylated TH. For the unphosphorylated TH forms, the inhibition constants are significantly lower, i.e., lower than 3.5 nM.

#### 2.3.7. Inhibition of TH by DOPA

Sura et al. [23] determined the binding of DOPA to human TH isoforms. For the unphosphorylated enzyme, the $K_{d}$ is reported to be between 2.7 and 4.5 μM. For phosphorylated TH, the $K_{d}$ values are slightly higher and lie in the range 4.0–7.4 μM. Unlike the binding of DA to phosphorylated and unphosphorylated TH, the binding of DOPA shows only minor changes in dependence of TH’s phosphorylation state on TH activity. Sura et al. [23] conclude that DOPA appears not to be important for regulation of TH in vivo, whereas TH inhibition by DA is.

#### 2.3.8. TH Turnover Number/$V_{max}$ for Tyr as Substrate

The turnover number of Tyr for the wild-type TH is reported to be 3.33 s${}^{-1}$≈ 200 min${}^{-1}$ [24]. Sura et al. [31] report a similar value for the Tyr turnover number of 3.03 s${}^{-1}$. We use the 200 min${}^{-1}$ value for TH’s turnover number and show how $V_{max}$ ($k_{2}$) in dependence of TH concentration influences the regulation of DOPA by DA.

#### 2.3.9. TH $K_{M}$ (Tyr) Values, $k_{3}$

BRENDA gives an overview of 16 $K_{M}$ values for human TH (using L-Tyrosine as substrate), ranging from 0.0081 to 0.166 mM. The values cover 11 mutant enzymes and five recombinant isoforms of TH. The isoform $K_{M}$ values are 0.011 mM [90] and 0.055, 0.066, 0.074, and 0.166 mM [91] with the average of 0.0744 mM = 74.4 μM. The $K_{M}$ values for the different mutant and wild-type enzymes are: 0.0081, 0.014, 0.015, 0.016, 0.019, 0.02, 0.025 [92], 0.017 [93], 0.039, 0.042, 0.043, 0.046 (’wild-type’), 0.066 [24], and 0.08 mM (’wild type’) [23]. Sura et al. [23] also provide an overview of $K_{M}$ values for different mutants and the wild-type enzyme. Fossbakk et al. [25] reports a $K_{M}$ of 5.1 ± 2.2 μM, while Morgenroth et al. [94] had reported earlier a $K_{M}$ for Tyr of 55.3 ± 4.2 μM similar to one of the above TH isoform values by Nasrin et al. [91]. For $k_{3}$, we use here the average value by Nasrin et al. [91] of 74.4 μM.

#### 2.3.10. TH-Mediated Conversion of DOPA

Haavik [34] reports a $K_{M}$ ($k_{16}$) for the DOPA oxidation by TH of 56 ± 12 μM. The rate of disappearance was found to be very low. In our calculations, we used a $V_{max}$ ($k_{15}$) for the removal of DOPA by TH to be the same as the $V_{max}$ ($k_{9}$) for TYR, i.e., 2.3 μM/min.

#### 2.3.11. Monoamine Oxidase (MAO) EC 1.4.3.4

Li et al. [95] studied the liver enzyme at 11 ${}^{\circ }$C. They determined the $K_{M}$ ($k_{19}$ in our model) to 111 μM and the $k_{cat}$ (turnover number) ($k_{19}$) to 0.249 s${}^{-1}$≈ 15 min${}^{-1}$. Making a rough estimate by using Van ’t Hoff’s rule that for each increase by 10 °C, the rate constants doubles, we estimate $k_{cat}$ roughly to 90 min${}^{-1}$ ( = 2 × 2 × 1.5 × 15 min${}^{-1}$). Assuming a MAO concentration of 1 μM, we obtain 90 μM/min for $V_{max}$ ($k_{18}$).

## 3. Results

### 3.1. Factors Influencing DOPA Homeostasis

Since experimental parameter values may vary depending on the studying conditions/methods, we investigated the system’s homeostatic behavior when certain parameter values are varied.

#### 3.1.1. $K_{M}$ of Dopamine Loading into Vesicles

As indicated in the derivation of $DOPA_{set}$ above, an important parameter for DOPA homeostasis is the $K_{M}$ value for the transport of DA into vesicles ($k_{8}$, Equation (8)). A necessary, but not sufficient condition for robust homeostasis, is that k8≪DA, i.e., the kinetics of DA loading should be zero-order or close to zero order with respect to DA. To illustrate how the zero-order condition influences DOPA homeostasis, Figure 6 shows the controller’s behavior toward ROS perturbations (oxidative stress) by applying step-wise increased $k_{14}$ values during three successive phases (phase 1: $k_{14}$ = 0 min${}^{-1}$; phase 2: $k_{14}$ = 0.01 min${}^{-1}$; phase 3: $k_{14}$ = 0.1 min${}^{-1}$) and by varying $k_{8}$ around the experimentally determined value of 0.29 μM [81].

The framed panel in Figure 6 shows the calculations using the experimentally determined value of $k_{8}$ = 0.29 μM, while in panels (a) and (b), the $k_{8}$’s are lower by two and one orders of magnitude, respectively. In panel (d), the $k_{8}$ is one order of magnitude higher than the value determined by Volz et al. [81]. Figure 6 clearly shows that the controller’s accuracy to defend $DOPA_{set}$ is dependent on $k_{8}$.

#### 3.1.2. Influence of Compensatory Flux $j_{2}$

While a low $K_{M}$ value for the DA loading into vesicles ($k_{8}$) is important for ensuring a good controller accuracy, a sufficiently high compensatory flux $j_{2}$
(14)$$j_{2}=\left(\frac{k_{2}\cdot Tyr}{k_{3}\left(1+\frac{DA}{k_{4}}\right)\cdot \left(1+\frac{DOPA}{k_{16}}\right)+Tyr}\right)\left(\frac{k_{12}}{k_{12}+Tyr}\right)\left(\frac{k_{20}}{k_{20}+DA_{ex}}\right)$$
is also required to allow a workable derepression by DA.

This is shown in Figure 7b where $k_{2}$ ($V_{max}$ of TH) is increased by one order of magnitude. Figure 7a shows the response before the increase in $k_{2}$. However, the improved controller accuracy comes at the expense of a slower response time and higher DA levels.

From Equation (14), we see that an increase in Tyr, for example by increasing the inflow of Tyr by $k_{1}$, can have negative effects on the controller, due to the Tyr inhibition term $k_{12}/\left(k_{12}+Tyr\right)$, which decreases $j_{2}$ with increasing Tyr concentrations. Taking Figure 6c as a reference and increasing the $k_{1}$ by one order of magnitude (from 10 to 100 μM/min), Tyr increases by approximately one order of magnitude, which leads to an 11% reduction of $j_{2}$ in phase 3 and a significant offset of DOPA from $DOPA_{set}$ (Figure 8).

#### 3.1.3. TH Inhibition by DA

Unphosphorylated TH is subject to a much stronger DA inhibition (lower inhibition constant) than phosphorylated TH. Since the reported values of the inhibition constants ($k_{4}$ in the model) vary significantly (see Material and Methods), we were interested in the feedback loops’s behavior when $k_{4}$ is varied. Figure 9 shows, in comparison with the varied $k_{8}$ and $k_{14}$ values from Figure 6, the corresponding results when $k_{4}$ is decreased by one order of magnitude. Clearly, a tighter inhibition of TH by DA results in a much more rapid response by the system, but due to a decreased compensatory flux, DOPA levels are below the controller’s set-point at larger $k_{8}$ values. In addition, DA steady-state levels are lower in comparison with Figure 6. As in Figure 6, the outlined panel c shows the results with $k_{8}$ having the value of 0.29 μM determined by Volz et al. [81].

#### 3.1.4. Levodopa Treatment

Since increasing oxidative perturbations $k_{14}$ leads to a decrease in DOPA and thereby to an additional decrease in DA, we investigated the effect when external DOPA is added, e.g., when medication by Levodopa is applied (described by rate constant $k_{22}$). Adding external DOPA to the system has two effects:(i)When applied such that the combined fluxes $j_{5}$ + $j_{11}$ + $j_{23}$ are lower than $j_{7}$ ($j_{7}$ is the rate of DA loading of vesicles, Figure 5), DOPA inflow helps to maintain DOPA homeostasis and slightly improves the performance of the controller/negative feedback. However, the improvement by DOPA addition is dependent on the controller accuracy (i.e., $k_{8}$ values). This is shown in Figure 10 where controller performances in absence of DOPA addition (panels a and b), and in its presence, (panels c and d) are compared for two different $k_{8}$ values. When $k_{8}$ is low and controller accuracy is high, DOPA addition improves controller performance and raises DA levels. In this case, all components of the model, including DA, are in a steady state.

(ii)However, if DOPA inflow results in the flux condition $j_{5}+j_{11}+j_{23}>j_{7}$, then the controller breaks down, and DA levels start to grow continuously with a DOPA steady-state level above $DOPA_{set}$. The controller tries to oppose the increased DOPA levels by downregulating the compensatory flux $j_{2}$ to zero with a continuous increase in DA. This behavior, also termed integral wind-up [57], is shown in Figure 11, when the DOPA inflow rate in phase 3 was increased to $k_{22}$ = 8 μM/min. DOPA steady-state levels are now entirely uncontrolled. Since the compensatory flux $j_{2}$ is practically zero, the steady-state level of DOPA is now solely determined by the DOPA inflow rate ($k_{22}$) and by the rates of DOPA removal.

As a result of the increasing DA levels during the wind-up, the rate of vesicular DA loading $j_{7}$ (Equation (9)) becomes zero-order ($k_{8}\ll DA$) with respect to DA, which has the effect that the vesicular concentration of DA ($DA_{ves}$) reaches its maximum steady-state value $k_{7}/\left(k_{17}+k_{23}\right)$:(15)$$DA_{ves}=\frac{k_{7}}{k_{17}+k_{23}}\left(\frac{DA}{k_{8}+DA}\right)=DA_{ves}^{max}\left(\frac{DA}{k_{8}+DA}\right)\;\;\overset{\mathrm{high}\;\mathrm{DA}}{\to }\;\;DA_{ves}^{max}$$

DOPA additions can restore DOPA levels when the activity of TH is low, such as in TH deficiency (THD) [96] or in DOPA-responsive dystonia (DRD) [97,98] patients. In the case that TH activity is not fully absent, DOPA additions can restore the homeostatic regulation of DOPA in presence of oxidative stress ($k_{14}$). This is indicated in Figure 12 where the amount of TH in phases 2 and 3 (both panels) was decreased by two-orders of magnitude to $k_{2}$ = $1\times 10^{2}\;\mu$M/min with $k_{14}$ = 0.1 μM/min. By comparison, phases 2 do not have any DOPA addition, while DOPA addition occurs in phase 3 ($k_{22}$ = 7 μM/min). The breakdown of the controller is seen in phase 2 showing low DOPA and DA levels due to the insufficient TH activity. As a reference, in phase 1, sufficient TH is present ($k_{2}$ = $1\times 10^{4}\;\mu$M/min), and oxidative stress is absent ($k_{14}$ = 0.0 μM/min). Panels a and b differ by controller accuracy $k_{8}$, showing that a lower $k_{8}$ (panel b) gives a better controller accuracy.

Since in the complete absence of TH activity the negative feedback is lacking, levodopa titrations need to be used to compensate for the missing levels of DOPA and DA.

#### 3.1.5. Robust DOPA Homeostasis Implies Maximum Vesicular DA Loading

As DA decreases in order to counteract for an outflow perturbation in DOPA, the vesicular DA concentration $DA_{ves}$ also decreases (Equation (15)). The accuracy of the DOPA controller, i.e., the value of $k_{8}$ (compared with Figure 6), then, according to Equation (15), determines how much the steady-state values of vesicle-loaded DA ($DA_{ves}$) change in relation to DA levels. Figure 13 shows $DA_{ves}$ steady-state levels (in terms of percentage of its maximum value $DA_{ves}^{max}$) in relation to DA concentrations and $k_{8}$. The black line 3 shows the $DA_{ves}^{max}-DA$ relationship when $k_{8}$ = 0.29 μM [81]. The other curves show the changes when $k_{8}$ is increased or decreased by one and two orders of magnitude.

Thus, a robust controller with low $k_{8}$ values keeps $DA_{ves}$ levels close at its maximum $k_{7}/\left(k_{17}+k_{23}\right)$ (Equation (15)) even when cytosolic DA levels change. Figure 14 shows the decrease in vesicular DA concentration in dependence of the changed controller accuracy $k_{8}$ and applied oxidative stress. The same perturbations by $k_{14}$ are applied as in Figure 6. Curves 1, 2, and 3 refer to the conditions in Figure 6a,c,d, respectively.

One may argue that maximum filled DA vesicles also occur when external DOPA is applied in excess, as indicated by Figure 11 and Equation (15). However, when DOPA is in excess, DA shows the wind-up and becomes uncontrollably high. The resulting high DA and DOPA levels can lead to increased and unfavorable oxidative stress by reactions with MAO/ROS leading to cell toxicity and neuropsychiatric disorders [30,45,99,100].

#### 3.1.6. Deteriorated DOPA Homeostasis by DA Removal/Auto-Oxidation

A removal of DA by MAO and by DA auto-oxidation disturbs DOPA homeostasis and may lead to the breakdown of the DOPA-DA control loop. Although the channeling complex may keep DOPA and DA protected against possible molecular attack by ROS, some “capturing” by ROS and/or MAO may occur.

Figure 15 shows how DA removal leads to controller breakdown. When α = 0, no DA removal occurs, and DOPA control by DA is operative. When MAO/auto-oxidation has access to DA (α = 1), DOPA homeostasis breaks down leading to high DOPA and low DA levels (Figure 15, dashed lines).

The corresponding decrease in $DA_{ves}$ levels is shown in Figure 16. In phase 3 when the perturbation $k_{14}$ is relative high ($k_{14}$ = 0.1 μM/min), there is a significant reduction in $DA_{ves}$ when DA oxidation is present (α=1), compared with the situation DA oxidation does not occur (α=0).

## 4. Discussion

### 4.1. On DOPA Regulation by DA Derepression in Cells

Using PC12 cells, Goldstein et al. [101] applied different MAO inhibitors and analyzed their influence on a variety of metabolites, including DA and DOPA. The study clearly shows the ”inverse” relationship between DA and DOPA: by applying increasing amounts of each inhibitor, DA levels rise (because less DA is oxidized by MAO), while DOPA levels correspondingly decline. Interestingly, for higher clorgyline concentrations (larger than 10 nM), the trend is reversed, i.e., DA levels decline with increasing inhibitor concentration and DOPA levels correspondingly rise. These results support the view of DOPA regulation by DA repression/derepression. However, we were not able to find the literature data which would answer the question to what extent the TH-DA negative feedback leads to more or less robust DOPA homeostasis. To address this, more specific tests are required, such as varying expression of TH or applying perturbations, which specifically would only reduce DOPA levels.

Thus, we analyzed the TH-DA negative feedback loop (Figure 5) from an ideal working perspective: under what conditions would an ideal control be possible, and which of the species within the negative feedback loop would be controlled? We see one kinetic and one structural argument which would favor ideal control.

The kinetic argument for an ideal robust control of DOPA is a zero-order uptake of DA into the vesicles. It implies, as seen in Figure 6, that DOPA is a regulated species and DA acts, by derepression kinetics, as the regulator. The experimental data by Volz et al. [81] for the $K_{M}$ ($k_{8}$ in the model) of the VMAT2 transporter indeed support this view, although the behavior of the controller when using the experimental $K_{M}$ value by Volz et al. only tangents an ideal DOPA control (Figure 6c). On the other hand, increased TH activity levels due to increased enzyme concentrations or a changed phosphorylation status can lead to an improved DOPA homeostasis (Figure 7).

The structural aspect is the indication that the enzymes TH, DCC, and the transporter VMAT2 appear to form a channeling complex [9,10,11]. The occurrence of metabolite channeling is well recognized [7] and is considered to be a central control mechanism of cellular metabolism [102]. There are two aspects of metabolic channeling which appear of importance in the context of DOPA regulation by DA. One is an effective DA filling mechanism of vesicles. The other is the minimization of environmental perturbations (for example, at increased oxidative stress) acting on DOPA and DA (Figure 15).

### 4.2. Why DOPA Homeostasis?

An argument suggested by Equation (15) is that under ideal controller conditions (low $k_{8}$), DOPA homeostasis allows an optimum DA filling of the vesicular stores, which seems to be of importance for a reliable translation of the electrical signal into a chemical one. In this respect, it is interesting that TH localization to VMAT2 is found to be regulated by its phosphorylation on Ser31 [12]. To what extent this regulation is also related to DA synthesis and DOPA homeostasis remains to be discovered. In addition, DOPA homeostasis could also be related to the regulation of flux control within a biosynthesis pathway originating from DOPA.

Figure 7 shows that TH is important for providing a sufficiently high compensatory flux $j_{2}$ for maintaining a functional DOPA controller. Since many cofactors are involved in the activity of TH, the dysregulation of these factors/genes has negative effects on DOPA control and DA levels. For example, the point mutation P301A and several others [25] show a decrease in TH’s activity $(V_{max}\left(Tyr\right)$) by two orders of magnitude, which certainly has an impact on DOPA homeostasis and the regulation of DA, as indicated above.

### 4.3. Oxidative Stress and Age

Reactive oxygen species are implicated in a variety of age-related diseases [103]. With regard to aging, DA and serotonin are those neurotransmitters that are mostly discussed. From early adulthood, DA levels decline steadily by around 10% per 10 years [104], which has been related to the increase in MAO activity with age [74]. Other factors that are implicated in the aging brain is the production of ROS by mitochondria [105]. ROS, which decrease DOPA levels, cause a reduction in DA, as the feedback loop tries to oppose the declining DOPA levels. When oxidative stress increases with age and removes DOPA, DA levels therefore also lead to a steady decrease. With an additional removal of DA by MAO, cytosolic and vesicular DA further decrease. As MAO produces hydrogen peroxide and radicals that may further decrease DOPA, a self-amplifying decrease in DA is expected to occur with age. On the other hand, there is a complex network of antioxidant molecules to keep an intracellular redox homeostasis [106], which needs to be considered in understanding the mechanisms of aging.

ATP is an important component which is required to maintain various homeostatic regulations, including protein (i.e., α-synuclein) homeostasis [107], the transport of DA into vesicles [10], and vesicle transport [5]. As ATP levels decline during aging [108], many of the homeostatic mechanisms decline with increased age.

### 4.4. Why Only DOPA Inflow Control?

Most of the homeostatic regulators in metabolism come in antagonistic pairs. Some examples are insulin and glucagon in blood sugar homeostasis or parathyroid hormone and calcitonin in blood calcium homeostasis (see the Supporting Material in [64] for descriptions of other controller pairs). Each individual controller in such a pair acts either as an inflow controller to allow minimum acceptable levels of the controlled metabolite or as an outflow controller to avoid that metabolite concentrations become too high [64]. As seen by the calculations above, the control of DOPA by DA opposes the removal of DOPA, for example by oxidative stress. The TH-DA negative feedback acts therefore as an inflow controller [64] to avoid that DOPA levels become too low. The reason for this is apparently that one needs DOPA for the making of DA, where the latter is needed in providing the chemical transmitter signal. However, this inflow control by DA does not oppose any DOPA levels above $DOPA_{set}$. This is seen in Figure 10b, when Levodopa is applied in such a high dose leading to the controller shutdown and DA wind-up. The absence of an outflow controller which would restrict maximum DOPA levels seems to indicate that higher DOPA levels can be better tolerated than lower ones.

### 4.5. Role of Other TH Regulators

The presence of Tyr inhibition on TH has in our calculations a clear influence on DOPA homeostasis by reducing the compensatory flux $j_{2}$ when Tyr increases. This is seen in Figure 8 when the Tyr concentration is increased by a larger $k_{1}$ value. However, this does not mean that robust homeostasis in DOPA cannot be achieved in the presence of Tyr inhibition of TH. Interestingly, calculations by Best et al. [27] indicate that the substrate inhibition of TH by Tyr can work as an attenuator, i.e., the Tyr inhibition of TH ’flattens’ the reaction rate catalyzed by TH and smooths $DA_{ves}$ concentration when Tyr levels fluctuate, for example, by the intake of meals.

## 5. Conclusions and Outlook

The results above suggest the view that DA regulates the homeostasis of DOPA by derepression. The critical parameter to achieve robust DOPA control is the $K_{M}$ ($k_{8}$) for the VMAT2 transporter moving DA into vesicles. The results by Volz et al. [81] for this $K_{M}$ (0.289 μM), and the value by Near ($5.4\times 10^{-3}\mu$M, for the binding of [2-3H] dihydrotetrabenazine to bovine striatal synaptic vesicles) [82], indicate that these transporter $K_{M}$s appear to be relative low and may work under zero-order conditions with respect to the uptaken compound. While $k_{8}$ is important for the accuracy of DOPA homeostasis (Figure 6), an increase in TH’s $V_{max}$ ($k_{2}$) is able to improve controller accuracy even when $k_{8}$ is relatively high (Figure 7). However, a sufficiently high compensatory flux $j_{2}$ is needed to maintain DOPA homeostasis. Factors which decrease $j_{2}$ may lead to a poorer controller performance, as for example an increase in TH inhibition by Tyr. The dephosphorylation of TH results in a stronger TH inhibition by DA, which leads to a more rapid and stronger controller response (Figure 9).

An argument for robust DOPA homeostasis is the observation that a zero-order kinetic loading of DA into vesicles not only results in robust DOPA control but also in a maximum DA loading of vesicles. This appears to be necessary in order to reliably transform an incoming train of electrical signals into a chemical transmitter message. Although we presently cannot answer whether the in vivo system shows robust DOPA control, we feel that our findings points to a possible “functional intention” of DOPA control during evolution. This regulatory feature of DA synthesis could be of interest for treatment procedures using MAO inhibitors or DOPA supplementation.

We have so far not included the (oscillatory) dynamics of repeated Ca${}^{2+}$ inflow, DA release into the synaptic cleft, and the influences on TH activity via auto-receptors and the re-entry of extracellular DA into the cell. Even under such oscillatory conditions, DOPA homeostasis may still be operative as calculations on related homeostatic controllers with derepression kinetics have shown [109,110]. We will study these oscillatory aspects in relation to DOPA/DA regulation in subsequent work.

## Figures and Tables

**Figure 1 ijms-22-12862-f001:**
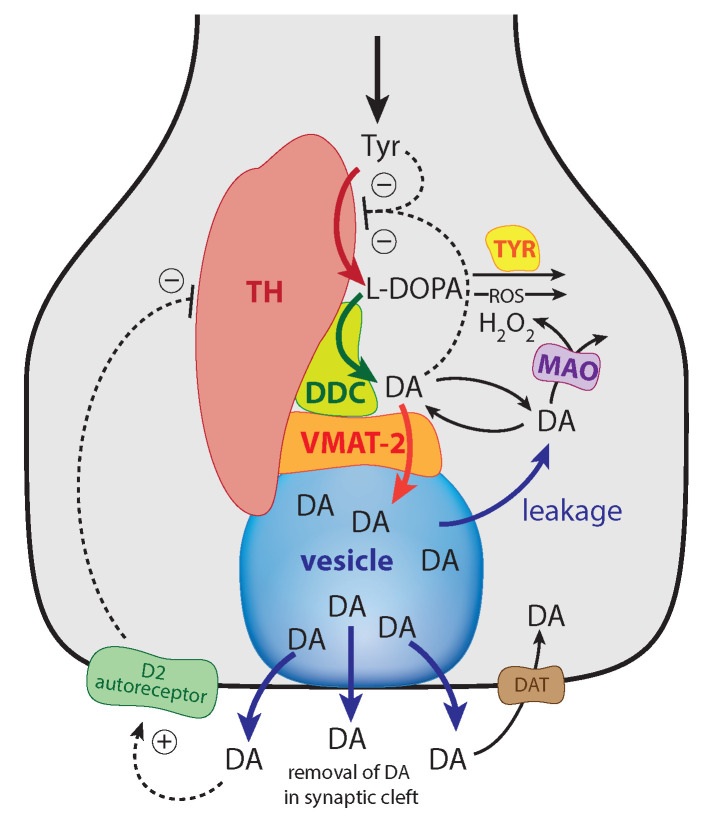
Overview of the synthesis and regulation of DA. There is evidence that DA is synthesized from tyrosine (Tyr) by a channeling mechanism with the enzymes TH, DDC, and the transporter VMAT2, forming a complex at the vesicular membrane [9,10,11]. The complex, whose details are presently not known, possibly minimizes the oxidations of DA and DOPA by MAO and reactive oxygen species (ROS), respectively [10]. It should however be noted that our analysis does not rely on the existence of such a protein complex. The synthesized DA is stored in synaptic vesicles for release. Significant leakage of transmitter molecules out from vesicles has been observed [16,17,18,19,20], and we tentatively assume that leakage also occurs in vivo. Upon arrival of an action, potential vesicles are emptied, and DA is released into the synaptic cleft. There, DA diffuses to DA-responsive target sites (postsynaptic neurons) or taken up by DA transporters (DAT) at neighboring dopaminergic terminals or metabolizing glial cells [10,14,21]. TH is inhibited both by DA [22,23] and its substrate Tyr [24,25]. There is also an inhibition of TH from extracellular DA via D2 autoreceptors that inhibits stimulatory cAMP/PKA phosphorylation of TH [26,27]. Tyrosinase (TYR) [28,29], converts DOPA to dopaquinone with the final formation of Neuromelanin [30]. Monoamine oxidase (MAO) forms hydrogen peroxide (a reactive oxygen species (ROS)) during the metabolization of DA. ROS are able to oxidize DOPA using a similar pathway as TYR, leading to neuromelanin (see [30] and references therein).

**Figure 2 ijms-22-12862-f002:**
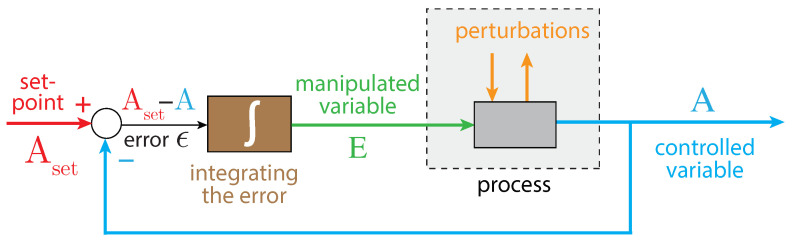
The principle of integral control in negative feedback regulation. Variable *A* is the controlled variable with set-point $A_{set}$. The error $\epsilon =A_{set}-A$ is integrated in time, which leads to the manipulated variable *E*, which can oppose perturbations affecting *A*. For step-wise perturbations, integral control leads to the precise regulation of *A* to $A_{set}$ [57].

**Figure 3 ijms-22-12862-f003:**
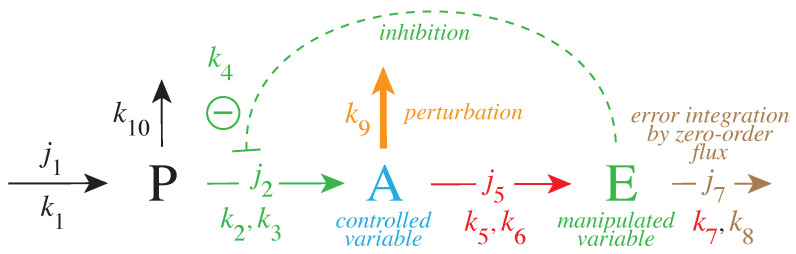
Negative feedback loop with robust homeostasis of controlled variable *A*. Same color code as in Figure 2 indicating the parameters which contribute to $A_{set}$ (red), the controlled variable *A* (blue) and the manipulated variable *E* (green). Outlined in brown is the parameter which leads to the integration of the error ϵ in Figure 2. The error integration is achieved when the manipulated variable *E* is removed by zero-order kinetics with respect to *E*, i.e., $k_{8}\ll E$ (see Equations (Equation 3)–(Equation 5) in main text). The pair of rate constants $k_{i}$, $k_{j}$ represent the respective $V_{max}$ and $K_{M}$ values of applied Michaelis–Menten kinetics. Compound *P* is the source of the compensatory flux $j_{2}$. $j_{2}$ needs to have a sufficiently high value in order to compensate for the step-wise perturbations $k_{9}$ which remove *A*.

**Figure 4 ijms-22-12862-f004:**
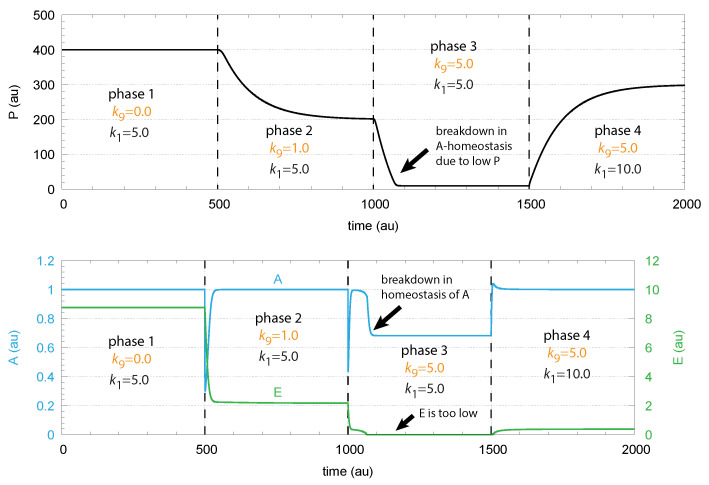
Demonstration of robust *A* homeostasis in response to step-wise changes of $k_{9}$. To keep *A* at its set-point at increased $k_{9}$ values, the concentration of *E* has to decrease by derepression under zero-order kinetics with respect to *E*. Upper panel: concentration of P as a function of time. Lower panel: concentrations of regulated species *A* (in blue) and controller species *E* (in green) as functions of time. For illustration purposes, the rate constants were chosen such, that $A_{set}$ = 1.0. Phase 1: Perturbation $k_{9}$ = 0.0. Rate constant $k_{1}$ represents the inflow of P, which is changed from 5.0 to 10.0 at the transition from phase 3 to phase 4. Phase 2: $k_{9}$ = 2.0; phases 3 and 4: $k_{9}$ = 5.0. Robust homeostasis in *A* breaks down in phase 3, because there is not sufficient *P* required to generate a sufficiently large compensatory flux $j_{2}$. When $k_{1}$ is increased in phase 4, homeostasis of *A* is reestablished by the increase in *P* and $j_{2}$. Other rate constants are: $k_{2}$ = 10.0, $k_{3}$ = 10.0, $k_{4}$ = 1.0, $k_{5}$ = 2.0, $k_{6}$ = 1.0, $k_{7}$ = 1.0, $k_{8}$ = $1\times 10^{-6}$, and $k_{10}$ = $1\times 10^{-2}$. Initial concentrations: $P_{0}$ = 400.0, $A_{0}$ = 1.0, and $E_{0}$ = 8.76. All values are in arbitrary units (au).

**Figure 5 ijms-22-12862-f005:**
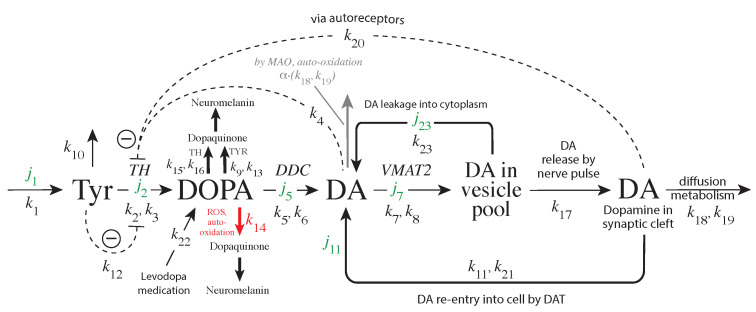
Reaction scheme of the TH-DA negative feedback model with added rate constants and fluxes. Not all DOPA and DA are necessarily bound to the channeling complex, but since the complex is not evenly distributed within the cytoplasm, both cytosolic DOPA and DA are considered to be located proximate to the complex. With increasing age and increased MAO levels [74] MAO may oxidize DA and perturb the feedback control of DOPA (path outlined in gray). In addition, both DOPA and DA are subject to auto-oxidation [75,76], which are included in the model by $k_{14}$ (together with ROS) and $\alpha (k_{18},\;k_{19})$ (together with MAO). As in Figure 3, pairs of rate constants ($k_{i}$, $k_{j}$) represent the respective ($V_{max}$, $K_{M}$) values of applied Michaelis–Menten kinetics. With increased MAO concentrations, oxidative stress and ROS also increase due to MAO-generated hydrogen peroxide. ROS can oxidize DOPA [30] (outlined in red). Cytosolic DA concentration is increased due to the re-entry of DA (flux $j_{11}$) and by vesicular leakage [16,17,18,19,20] (flux $j_{23}$). Inflow of Ca${}^{2+}$ by neuronal stimulation lead to release of vesicular DA into the synaptic cleft. Clearance of DA in the synaptic cleft occurs by diffusion, glial cell metabolization [14,21], or uptake of DA by DAT [77,78].

**Figure 6 ijms-22-12862-f006:**
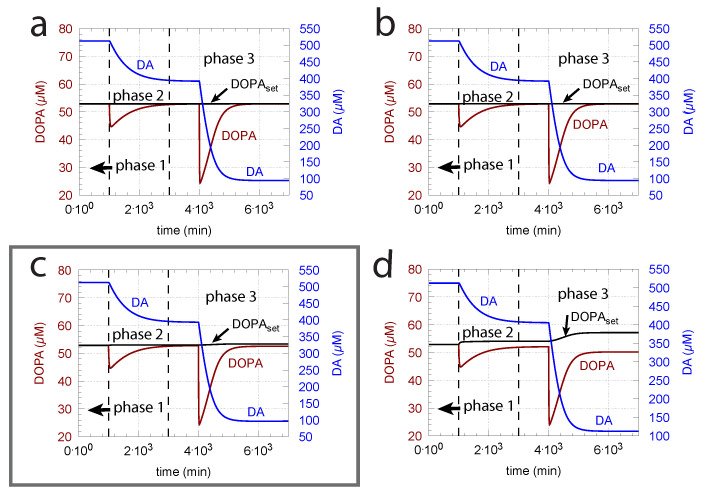
The $K_{M}$ ( = $k_{8}$) of the transport of DA into vesicles has a major influence on the controller accuracy for DOPA homeostasis toward step-wise increased perturbations by oxidative stress ($k_{14}$). (**a**) $k_{8}$ = $2.9\times 10^{-3}\;\mu$M; (**b**) $k_{8}$ = $2.9\times 10^{-2}\;\mu$M; (**c**) $k_{8}$ = $2.9\times 10^{-1}\;\mu$M; (**d**) 2.9 μM. Other rate constants (**a**–**d**): $k_{1}$ = 10 μM/min, $k_{2}$ = $1\times 10^{4}$
μM/min, $k_{3}$ = 74.4 μM, $k_{4}$ = 6 μM, $k_{5}$ = 10.0 μM/min, $k_{6}$ = 346 μM, $k_{7}$ = 6.0 μM/min, $k_{9}$ = 2.3 μM/min, $k_{10}$ = 0.025 min${}^{-1}$, $k_{11}$ = 1 μM/min, $k_{12}$ = 50 μM, $k_{13}$ = 437 μM, $k_{14}$ phase 1: 0.0 min${}^{-1}$, $k_{14}$ phase 2: 0.01 min${}^{-1}$, $k_{14}$ phase 3: 0.1 min${}^{-1}$, $k_{15}$ = 2.3 μM/min, $k_{16}$ = 56 μM, $k_{17}$ = 0.01 min${}^{-1}$, $k_{18}$ = 90 μM/min, $k_{19}$ = 111 μM, $k_{20}$ = 0.1 μM, $k_{21}$ = $1\times 10^{-4}\;\mu$M, $k_{22}$ = 0.0 min${}^{-1}$, $k_{23}$ = 0.0158 min${}^{-1}$. Initial concentrations, (**a**–**d**): Tyr = 323.2 μM, DOPA = 52.9 μM, DA = 512.5 μM, $DA_{ves}$ = 232.6 μM, $DA_{ex}$ = 1.7 μM. Framed panel c shows the calculations with the experimentally determined value [81] of $k_{8}$.

**Figure 7 ijms-22-12862-f007:**
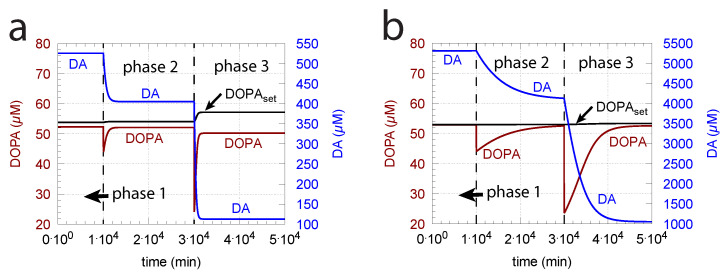
Increase in $V_{max}$ ($k_{2}$) improves the accuracy of DOPA homeostasis but leads to a slower response of the controller. Panel (**a**) is a recalculation of Figure 6d. Panel (**b**) shows the controller’s response when $k_{2}$ has increased from $k_{2}$ = $1\times 10^{4}$
μ to $k_{2}$ = $1\times 10^{5}$
μM/min. Other rate constants as in Figure 6d. Initial concentrations, panel (**b**): Tyr = 323.1 μM, DOPA = 53.0 μM, DA = $5.3\;\times \;10^{3}\;\mu$M, $DA_{ves}$ = 232.4 μM, $DA_{ex}$ = 1.7 μM. Note that the controller’s accuracy can be further improved by having $k_{8}$ values lower than 2.9 μM.

**Figure 8 ijms-22-12862-f008:**
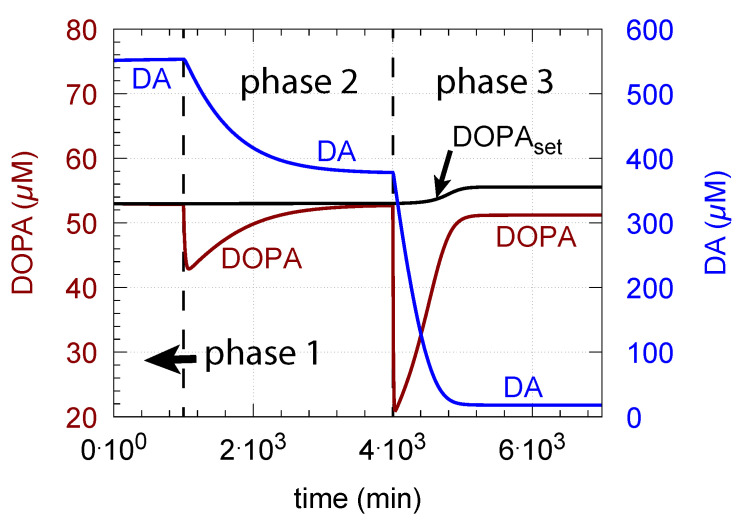
Increase in $k_{1}$ leads to a poorer homeostatic regulation of DOPA due to TH inhibition by increased Tyr levels. The graph shows DOPA and DA concentrations when $k_{1}$ = 100 μM/min. Other rate constant values as in Figure 6c. Initial concentrations: Tyr = $3.7\times 10^{3}\;\mu$M, DOPA = 51.2 μM, DA = 18.1 μM, $DA_{ves}$ = 228.9 μM, $DA_{ex}$ = 1.6 μM.

**Figure 9 ijms-22-12862-f009:**
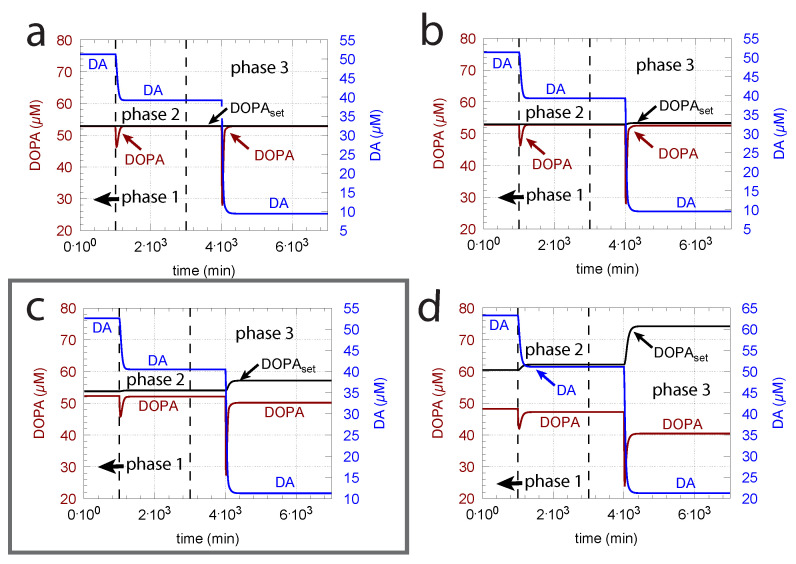
Stronger inhibition of TH by DA (lower $k_{4}$), for example by dephosphorylation [22], leads to more rapid response kinetics (in comparison with Figure 6). However, the stronger inhibition also leads to lower DA steady-state levels and to a larger offset of DOPA from its set-point. Rate constants as in Figure 6, except for $k_{4}$, which was reduced by one order of magnitude to 0.6 μM. Initial concentrations: (**a**) Tyr = 323.2 μM, DOPA = 52.9 μM, DA = 51.3 μM, $DA_{ves}$ = 232.5 μM, $DA_{ex}$ = 1.7 μM; (**b**) Tyr = 323.3 μM, DOPA = 52.8 μM, DA = 51.4 μM, $DA_{ves}$ = 232.4 μM, $DA_{ex}$ = 1.7 μM; (**c**) Tyr = 324.0 μM, DOPA = 52.3 μM, DA = 52.6 μM, $DA_{ves}$ = 231.3 μM, $DA_{ex}$ = 1.6 μM; (**d**) Tyr = 329.3 μM, DOPA = 48.2 μM, DA = 63.3 μM, $DA_{ves}$ = 222.4 μM, $DA_{ex}$ = 1.5 μM. Framed panel c shows the calculations with $k_{8}$ = 0.29 μM [81].

**Figure 10 ijms-22-12862-f010:**
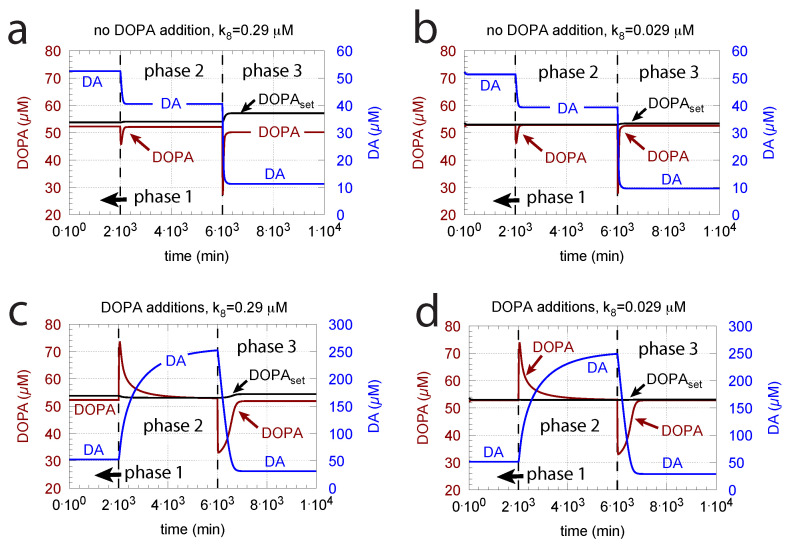
DOPA addition can improve DOPA homeostasis when $j_{5}+j_{11}+j_{23}<j_{7}$. Panels a and b show reference calculations with two different $k_{8}$ values (representing different controller accuracies) when DOPA is not added ($k_{22}$ = 0 μM/min). Panels c and d show the effect of DOPA additions. For all panels, (**a**–**d**): phase 1, no DOPA addition; phase 2, $k_{22}$ = 2 μM/min addition; phase 3, $k_{22}$ = 4 μM/min addition. Other rate constant values are as in Figure 9. Initial concentrations in panels a and c are as in Figure 9c. Initial concentrations in panels b and d are as in Figure 9b.

**Figure 11 ijms-22-12862-f011:**
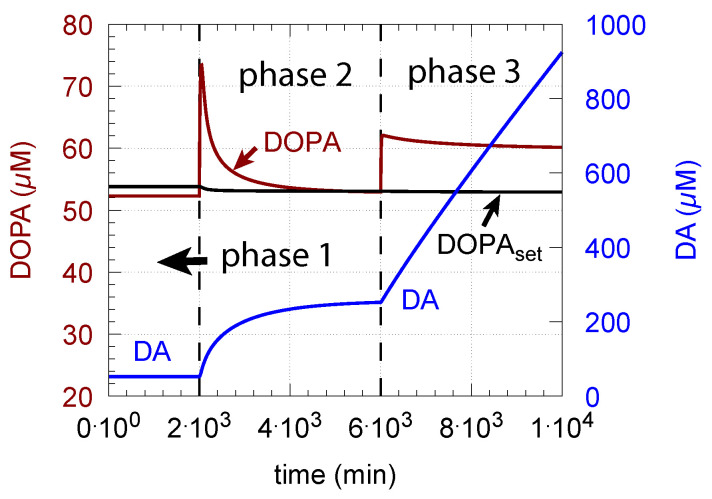
DA integral wind-up occurs when, here in phase 3, $k_{22}$ is set to 8 μM/min, which results in the condition $j_{5}+j_{11}+j_{23}>j_{7}$. Other rate constant values and initial concentrations as for Figure 10c.

**Figure 12 ijms-22-12862-f012:**
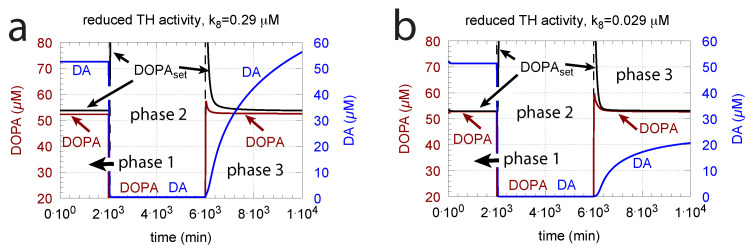
Re-establishing robust DOPA homeostasis at reduced TH activity levels by DOPA addition. In both panels: phase 1 has a normal TH activity ($k_{2}$ = $1\times 10^{4}\;\mu$M/min) and no oxidative stress ($k_{14}$ = 0.0 μM/min); in phase 2, oxidative stress is present ($k_{14}$ = 0.1 μM/min), and TH activity level is reduced by two orders of magnitude ($k_{2}$ = $1\times 10^{2}\;\mu$M/min). As a result, DOPA homeostasis breaks down in phase 2 with low DOPA and DA levels. In phase 3, DOPA is added ($k_{22}$ = 7 μM/min) in the presence of low TH and oxidative stress as in phase 2. In phase 3, DOPA homeostasis is restored. Panel (**a**): $k_{8}$ = 0.29 μM; panel (**b**): $k_{8}$ = 0.029 μM. Other rate constant values and initial concentrations, as in Figure 10a.

**Figure 13 ijms-22-12862-f013:**
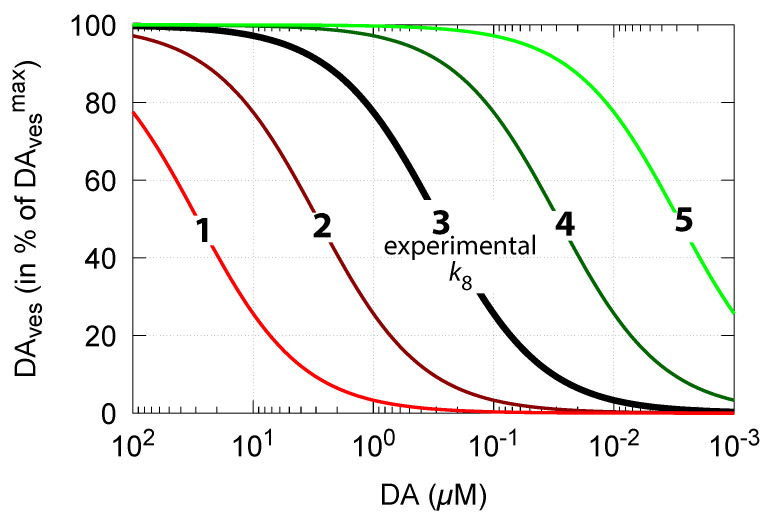
Calculated vesicular steady-state DA concentrations $DA_{ves}$ as a function of cytosolic DA levels and DOPA controller accuracy $k_{8}$ (Equation (15)). At high controller accuracy (low $k_{8}$, line no. 5), $DA_{ves}$ levels are less affected when DA concentrations decrease. At low controller accuracy (high $k_{8}$, line no. 1), $DA_{ves}$ levels decrease rapidly with increased DOPA perturbations and with decreasing DA concentrations. $k_{8}$ values used: **1**, 29 μM; **2**, 2.9 μM; **3**, 0.29 μM; **4**, 0.029 μM; and **5**, 0.0029 μM.

**Figure 14 ijms-22-12862-f014:**
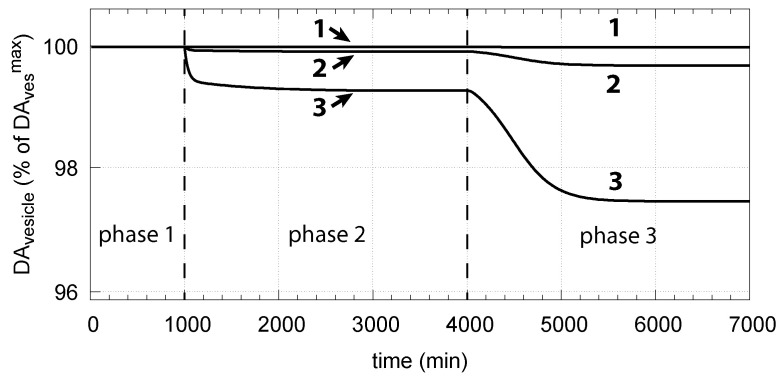
Vesicular DA concentration as a function of perturbation $k_{14}$ and controller accuracy $k_{8}$. The step-wise changes in $k_{14}$ for phases 2 and 3, as well as the other rate constants and initial concentrations, are the same as in Figure 6. Curves 1, 2, and 3 refer to the conditions of Figure 6a,c,d, respectively.

**Figure 15 ijms-22-12862-f015:**
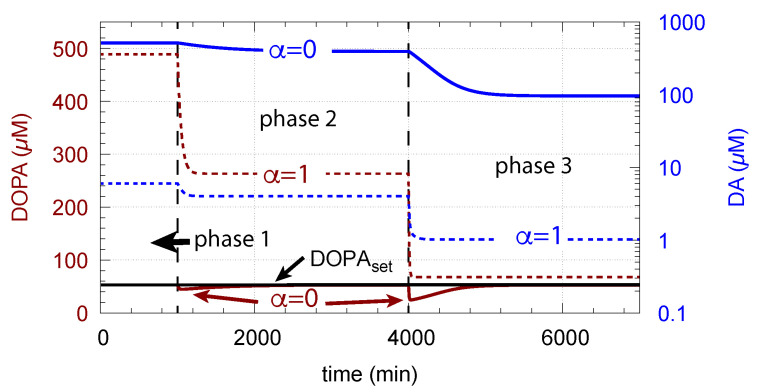
Deteriorated DOPA homeostasis when DA is subject to removal by MAO/auto-oxidation. Solid lines show the performance when α = 0 (Figure 6c). Dashed lines show the concentrations when α = 1 and DA is removed. Rate constant values as for Figure 6c with $k_{18}$ = 90 μM/min and $k_{19}$ = 111 μM. Initial concentrations (dashed curves, α = 1): Tyr = 210.7 μM, DOPA = 518.3 μM, DA = 1.705 μM, $DA_{ves}$ = 149.7 μM, $DA_{ex}$ = 0.62 μM. Red and blue lines show DOPA and DA concentrations, respectively.

**Figure 16 ijms-22-12862-f016:**
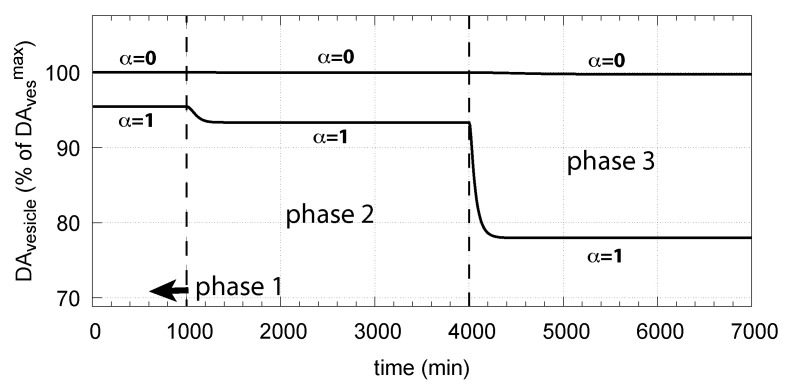
Effect of cytosolic DA oxidation/auto-oxidation on vesicular DA levels. α = 0 indicates the results when using the conditions as in Figure 6c, i.e., no DA removal occurs. α = 1 allows oxidation of DA to occur. Same rate parameters as in Figure 6c are applied. Initial concentrations for α = 1 are the same as in Figure 15.

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
