# Peer review of "DOPA Homeostasis by Dopamine: A Control-Theoretic View"

_ijms, 2021, doi:10.3390/ijms222312862_

Round 1

Reviewer 1 Report

The article of Kleppe et al looks at the regulation of the biochemical activities operating in the terminals of dopaminergic neurons and proposes the idea of “L-DOPA homeostasis » upon simulation. Briefly, it is known that the enzymatic pathway linking the transformation of tyrosine by tyrosine hydroxylase (TH) to the entry of the final product dopamine (DA) into the vesicles of exocytosis is extremely regulated, and complex. The intermediate compound, L-DOPA has not made the object of specific attention other than being the direct precursor of DA. Using different values cautiously selected for each enzymatic/transporter step, considering also the existence of numerous perturbations in the schemas, they address the possibility of the existence of L-DOPA homeostasis. They also simulated the outcome of the full pathway when exogenous L-DOPA is entered into the equations. The authors discuss the relevance of L-DOPAset in aging, oxidative metabolism and so on.

The idea is interesting, has obviously to be considered (maybe for the other monoaminergic systems), but I’m not sure that it suits well for DA in the striatum. At least, we can regret that the authors did not discuss experimental data (possibly pharmacology) in the light of their claim. It is the main drawback to my point of view, but I was also not totally confident in some statements/choices made by the authors, and it turns out that it looks critical.

  1. The authors should find some experimental data to discuss their finding. A lot of drugs would dramatically act on the equilibrium including inhibitors of monoamine oxidases, inhibitors of DDC, TH, amphetamine, competitor of L-DOPA like 5-HTP etc… Thus, although I found the idea pertinent, I’m rather confused without any experimental data to support the claims.
  2. The authors insisted several times on the fact that k8, the rate of disappearance of DA into the vesicles of exocytosis had strong importance in “DOPA homeostasis”. My concern is “k-8”. It is well known that there is a dynamic transport in both direction, and the raise in intracellular metabolism for catecholamines (maybe also for serotonin) is rather due to leaks from the vesicles, via VMAT2 (see for instance Eisenhofer, Kopin and Goldstein in Pharmacological reviews, 2004). In the present simulation, the authors presuppose that there is only one direction. I could understand the idea in noradrenergic neurons where DA would enter the vesicles via VMAT2, and won’t contribute that much in leaks because it is highly transformed into noradrenaline. Please discuss this point if it has importance.
  3. Another phenomenon has not been considered as a perturbation: the auto-oxidation of DA and L-DOPA.
  4. The authors could pay attention to abbreviations, notably dopamine (often not abbreviated).
  5. Figure 1 could be modified. First, the unidirectional arrow from the cytosol to the vesicles is open to discussion. Second, the product of DA through MAO is not HVA. It is not even DOPAC. Third, HVA, which should be removed, is not in the legend. Fourth, what are doing the MAOs in the synaptic cleft? At best, please justify it with appropriate references, but they are bound to mitochondrial membranes (see BRENDA).
  6. What about a system where there is no TH in case of exogenous L-DOPA? Or the role of serotonergic neurons in the mechanism of action of L-DOPA. Is L-DOPA regulated in serotonergic neurons beyond all the perturbations including DDC, auto-oxidation and so on?

Author Response

Please see attached file "reply_reviewer1.pdf"

Reviewer 2 Report

The authors describe the synthesis of a control system with TH-DA negative feedback loop for a robust DOPA regulation. This system is characterized of close to zero-order kinetics of the vesicular DA loading with a channelling mechanism of the involved enzymes.

My remarks are the following.

  1. This remark relates to the connection between the structures shown in Figure 2 and Figure 3. In Figure 2, I see a subtractor (Aset–A=e). But I don’t see subtraction in Figure 3. Blue colored A is followed by red colored variables j5, k5, k6. Then, E is set. This order is strange. To my mind, a subtractor is required for showing.
  2. To define parameters (variables) depicted in Figure 3, the optimization problem is to be solved. What is the mathematical form of this optimization problem and what criteria is used?

Author Response

Please, see attached file "reply_reviewer2.pdf".

Round 2

Reviewer 1 Report

The article of Kleppe et al looks at the regulation of the biochemical activities operating in the terminals of dopaminergic neurons and proposes the idea of “L-DOPA homeostasis » upon simulation. Briefly, it is known that the enzymatic pathway linking the transformation of tyrosine by tyrosine hydroxylase (TH) to the entry of the final product dopamine (DA) into the vesicles of exocytosis is extremely regulated, and complex. The intermediate compound, L-DOPA has not made the object of specific attention other than being the direct precursor of DA. Using different values cautiously selected for each enzymatic/transporter step, considering also the existence of numerous perturbations in the schemas, they address the possibility of the existence of L-DOPA homeostasis. They also simulated the outcome of the full pathway when exogenous L-DOPA is entered into the equations. The authors discuss the relevance of L-DOPAset in aging, oxidative metabolism and so on.

It is a revised version. The comments have been seriously, very seriously taken into account, and I do believe that the model has been implemented, and ameliorated. The authors have added some data by Goldstein et al recalling us that the effects of MAO inhibitors have always been a nightmare to interpret and the idea that L-DOPA might be a more important homeostatic variable than previously thought could be part of the nightmare.

To be frank with the authors, I recently completed a collaboration in which I measured the tissue levels of serotonin in mice receiving pCPA, an inhibitor of tryptophan hydroxylase. In the chromatograms, there was a peak corresponding to the precursor of serotonin, i.e. 5-HTP (which has very low endogenous concentrations, like L-DOPA and even less). There was a strong decrease in serotonin and its metabolite, but the levels of 5-HTP were only slightly, not significantly diminished. I came to the conclusion that the peak of 5-HTP could be something else (always possible), or... that 5-HTP could be a more important integrator than previously thought. And then the article of the authors came on my desk. So, continue...

In order to complete this, maybe the authors could look at some data with alpha-methyl-para-tyrosine, but I'm suspecting that no one could follow the endogenous levels of L-DOPA. A last note for the authors: these effects (control of endogenous level of L-DOPA) might be region-dependent because there is a huge heterogeneity of dopaminergic terminal functions in the brain (at least when we look at the metabolism of dopamine across the brain).